# Minor changes in the incidence of primary and secondary involuntary childlessness across birth cohorts 1916 to 1975, but major differences in treatment success

**Finn Egil Skjeldestad**⬤*

Research Group Epidemiology of Chronic Diseases, Institute of Community Medicine, UiT the Arctic University of Norway, Tromsø, Norway

* eskjelde@online.no

**Data Availability Statement:** All relevant data are within the paper and its Supporting Information files.

## Abstract

There have been tremendous advances in assisted reproductive technologies (ARTs) over the past 50 years. The present study assessed infertility outcomes among women of reproductive age during this period. The seventh survey of the Tromsø Study (Tromsø7, 2015–16) recruited Tromsø residents aged 40–98 years. The questionnaire collected information on sociodemographics and infertility, as well as data from a wide range of validated health questionnaires. Primary involuntary childlessness was defined as reporting one or more of the following: the clinical definition of infertility (i.e., infertility period of >1 year), infertility examination, use of ART, and/or the birth of a child conceived during ART. Women with secondary involuntary childlessness were those who reported infertility experience and had least one naturally conceived child. Parous women without infertility experience were classified as fertile, and nulliparous women without infertility experience as voluntarily childless. The main exposure was birth cohort (1916–35, aged 80–98 years; 1936–45, aged 70–79 years; 1946–55, aged 60–69 years; 1956–65, aged 50–59 years; 1966–75, aged 40–49 years). The incidence of primary involuntary childlessness was significantly higher in the 1956–75 cohort (6.0%; 95% CI: 5.4–6.6) than the 1916–55 cohort (3.7%; 95% CI: 3.2–4.3). The incidence of secondary involuntary childlessness was higher than that of primary involuntary childlessness across all birth cohorts and was highest for the 1966–75 cohort (10%), with no differences observed across the other birth cohorts (6–7%). An increasing proportion of women from the oldest to the youngest birth cohorts reported infertility examination and ART. ART success increased substantially with time, reaching 58% for primary and 46% for secondary infertility in the 1966–75 cohort. Voluntarily childless women comprised 5–6% of the 1916–55 cohort and 9–10% of the 1956–75 cohort. There were minor differences in the incidence of primary and secondary involuntary childlessness across the 1916–75 cohorts. Advances in ART over the past 50 years comprised 2.0% and 3.3% of population growth in the 1956–65 and 1966–75 cohorts, respectively: a remarkable achievement.

**Funding:** The author received no specific funding for this work.

**Competing interests:** The author has declared that no competing interests exist.

**Abbreviations:** ART, assisted reproductive technology; IVF, in vitro fertilization; CI, Confidence interval; OR, Odds ratio.

## Introduction

In ancient times, humans believed that fertility gods could assist with childlessness through rituals and offerings [1]. Four thousand years ago, the Egyptians were prescribing incense, fresh oil, dates, and beer to promote conception [2]. In the late 17th century, Leeuwenhoek, the constructor of the first microscope, identified sperm and sperm motility [3], but insemination was not introduced until the 1960s [4]. Indeed, over the last 50 years, there have been outstanding advancements in assisted reproductive technology (ART). During the 1960s and early 1970s, scientists made major developments in cell cultures; cell lines; the extraction of human gonadotropins from human menopausal urine and the understanding of their physiological function; advances in bioassays, purification processes, washing, freezing preservation, and thawing of semen; and the introduction of intrauterine insemination. These processes were continuously assessed and perfected, and finally resulted in the first child born by in vitro fertilization (IVF) in 1978 [4]; in 1984, the first child was born by IVF in Norway [5]. Norway was also the first country in the world to implement regulations regarding "artificial" fertilization in 1987 [6].

Researchers around the world continue to refine and optimize ovarian stimulation; egg retrieval; sperm preparation; embryo culture, evaluation, selection, transfer, freezing, preservation, and thawing; and endometrial receptivity. Today, we take infertility treatment, and its success, for granted.

The present study assessed infertility outcomes among women of reproductive age during the last 50 years of advancements in ART.

## Material and methods

The Tromsø Study [7, 8] is a large population-based survey that invites all Tromsø residents aged 40–98 years identified in the Central Population Register of Norway to participate in health surveys at timely intervals. Since 1974, seven surveys have been organized. The questionnaire collected sociodemographic and infertility information, as well as data from a wide range of validated health questionnaires, physical examinations, and biological materials. This analysis is based on data from women who participated in the 7th survey of the Tromsø Study (Tromsø7, completed in 2015–16).

## Variables

The main exposure was defined as birth cohort, categorized as the 1916–35 (aged 80–98 years), 1936–45 (aged 70–79 years), 1946–55 (aged 60–69 years), 1956–65 (aged 50–59 years), and 1966–75 (aged 40–49 years) cohorts. Marital status was categorized as unmarried, married, widowed, divorced, and separated. Married women who did not respond to the item "living with a spouse/partner (yes/no)", were categorized as living with a spouse/partner. Educational level was divided into primary (≤9 years of schooling), vocational/upper secondary (10–12 years of schooling), college/university <4 years (13–15 years of schooling), or college/university ≥4 years (≥16 years of schooling).

Infertility experience was defined as reporting one or more of the following: the clinical definition of infertility (i.e., infertility period of >1 year), infertility examination, use of ART, and/or the birth of a child conceived during ART. Women reported on the clinical definition of infertility through the question: "Have you tried to conceive for more than 1 year without succeeding in becoming pregnant?" (yes/no). Women were categorized as having had an infertility examination if they reported a "cause" of infertility (male infertility, female infertility, or both). Use of ART was assessed through five questions, asking if the woman had ever received any infertility treatment, ovulation stimulation, artificial insemination with a partner, artificial

insemination with a donor, or IVF/intracytoplasmic sperm injection. Women who did not reply to the question on the clinical definition of infertility, but reported use of ART or confirmed the birth of a child/children conceived during ART, whether in Norway or abroad, were recoded as having had an infertility period of more than 1 year.

Whereas births are reported quite accurately in surveys, ectopic pregnancies, and spontaneous and induced abortions are not as reliably reported [9, 10]. For these reasons, the Tromsø Study collected information on history of births, not pregnancies. Therefore, births were used in this study to determine fertility status [9].

Women who reported infertility experience and had no children/the same number of children as the number conceived during ART, were classified as having primary infertility/involuntary childlessness. Women with secondary infertility/involuntary childlessness were those who reported infertility experience and had at least one naturally conceived child. Parous women without infertility experience were classified as fertile, and nulliparous women without infertility experience as voluntarily childless. Women with primary and secondary involuntary childlessness were further divided into categories according to the services they received: no infertility examination and no ART, infertility examination only, or infertility examination and ART.

Data was taken primarily from Tromsø7 questionnaires. However, 43% (n = 4773) of the women also participated in Tromsø6 (2007–08; age 32–90 years). Therefore, we performed consistency analyses for these participants for the variables educational level, number of children, year of delivery, clinical infertility, and "cause" of infertility. When available, we replaced missing values in the Tromsø7 dataset with values from Tromsø6; in a few cases of inconsistency, the highest/most severe value was used. The fertility status of women (n = 15) who were pregnant at the time of Tromsø7 did not take into account the ongoing pregnancy [9].

Statistical analyses were done using IBM SPSS version 26.0 with Chi-square test and logistic regression. P-values <0.05 were considered statistically significant.

The Regional Committee for Medical and Health Research Ethics North (case no. 54329) and The Norwegian Centre for Research Data (project no. 298477) reviewed the protocol before study start. All participants gave written informed consent.

## Results

Of the 16 539 invited women, 11 064 (66.9%) were eligible for inclusion in the analysis. The response rate ranged from 65% to nearly 75% for the 1966–75 and 1946–55 cohorts, respectively, but it was remarkably lower (33.7%) for the 1916–35 cohort (Table 1).

A larger proportion of women in the younger birth cohorts were unmarried and living with a spouse/partner ($x^2$-trend, p<0.001) (Table 1). These birth cohorts also had significantly higher proportions of women with a college/university education (both <4 and ≥4 years) than did the older birth cohorts ($x^2$-trend, p<0.001) (Table 1). Overall, including children conceived during ART, a higher proportion of women in the 1916–35 cohort had three or more children, while women born after 1935 more often had one or two children (Table 1).

Less than 1% of included women had unknown fertility status. For further analyses, we recoded women with unknown fertility status and no children, and unknown fertility status and one or two children, to voluntarily childless women and fertile women, respectively (Table 2). The prevalence of primary infertility/involuntary childlessness was significantly higher among women born after 1955 (6.0%; 95% confidence interval (CI): 5.4–6.6) than among those from the 1916–55 cohorts (3.7%; 95% CI: 3.2–4.3) (Table 2). Similarly, the prevalence of secondary infertility/involuntary childlessness was highest among the youngest birth cohort (1966–75 cohort, women aged 40–49 years, 10.0%; 95% CI: 9.0–11.0), with no

**Table 1. Invitations, response rates, participation rates, and study population characteristics by birth cohort.**

| | Birth cohorts | | | | |
|---|---|---|---|---|---|
| | 1966–75 | 1956–65 | 1946–55 | 1936–45 | 1916–35 |
| Age (years) | 40–49 | 50–59 | 60–69 | 70–79 | 80–98 |
| Invited | N = 5195 | N = 4534 | N = 3586 | N = 2001 | N = 1223 |
| | % | % | % | % | % |
| Response rate | 65.0 | 71.5 | 74.6 | 68.0 | 33.7 |
| Participated | N = 3373 | N = 3244 | N = 2675 | N = 1360 | N = 412 |
| | % | % | % | % | % |
| Marital status | | | | | |
| Unmarried | 38.2 | 26.2 | 11.1 | 4.6 | 2.4 |
| Married | 49.1 | 51.8 | 58.2 | 56.3 | 28.2 |
| Widowed | 0.7 | 2.2 | 7.6 | 22.5 | 60.4 |
| Divorced | 10.2 | 17.9 | 22.2 | 16.3 | 8.7 |
| Separated | 1.8 | 1.8 | 0.9 | 0.4 | 0.2 |
| Living with a spouse/partner (yes) | 75.4 | 70.7 | 68.1 | 59.7 | 29.6 |
| Educational level | | | | | |
| Unknown | 0.6 | 0.6 | 0.8 | 0.8 | 1.9 |
| Primary | 7.8 | 17.0 | 33.0 | 53.2 | 66.3 |
| Vocational/upper secondary | 23.4 | 27.9 | 26.5 | 22.1 | 20.1 |
| College/university <4 years | 21.4 | 19.5 | 14.5 | 10.7 | 6.8 |
| College/university ≥4 years | 46.9 | 34.9 | 25.2 | 13.2 | 4.9 |
| Number of children | | | | | |
| 0 | 12.7 | 13.9 | 9.8 | 6.5 | 9.7 |
| 1 | 14.7 | 16.2 | 14.8 | 9.7 | 6.8 |
| 2 | 43.9 | 40.5 | 43.3 | 35.2 | 25.7 |
| ≥3 | 28.7 | 29.4 | 32.0 | 48.6 | 57.8 |

differences across the other birth cohorts (Table 2). The proportion of voluntarily childless women was significantly higher in the 1956–65 and 1966–75 cohorts (9.3%; 95% CI: 8.6–10.0) than in women born before 1956 (5.6%; 95% CI: 4.9–6.3) (Table 2).

The proportion of women with primary and secondary infertility was evenly distributed across birth cohorts ($x^2$-trend = 0.29) (overall distribution 39% versus 61%) (Table 3, upper panel). Among women with primary infertility, nearly 80% in the youngest birth cohort

**Table 2. Fertility status by birth cohort.**

| | Birth cohorts | | | | |
|---|---|---|---|---|---|
| | 1966–75 | 1956–65 | 1946–55 | 1936–45 | 1916–35 |
| Age (years) | 40–49 | 50–59 | 60–69 | 70–79 | 80–98 |
| | N = 3373 | N = 3244 | N = 2675 | N = 1360 | N = 412 |
| | % | % | % | % | % |
| Fertility status | | | | | |
| Unknown, no children | 0.1 | 0.4 | 0.6 | 0.4 | 0.5 |
| Unknown, 1–2 children | 0.4 | 0.4 | 0.3 | 0.6 | 0.2 |
| Childless | 8.9 | 9.2 | 5.8 | 3.5 | 5.1 |
| Fertile, 1–15 children | 74.4 | 76.4 | 82.5 | 86.0 | 84.0 |
| Primary infertility | 6.1 | 5.9 | 4.1 | 2.6 | 4.1 |
| Secondary infertility | 10.0 | 7.6 | 6.6 | 6.8 | 6.1 |

**Table 3. Proportion of women with primary infertility or secondary infertility, and women who received fertility examination and/or assisted reproductive technology (ART) by type of infertility (%).**

| | Birth cohorts | | | | |
|---|---|---|---|---|---|
| | 1966–75 | 1956–65 | 1946–55 | 1936–45 | 1916–35 |
| Age (years) | 40–49 | 50–59 | 60–69 | 70–79 | 80–98 |
| | N = 544 | N = 440 | N = 287 | N = 129 | N = 42 |
| Type of infertility | % | % | % | % | % |
| Primary infertility | 37.9 | 43.6 | 38.7 | 27.9 | 40.5 |
| Secondary infertility | 62.1 | 56.4 | 61.3 | 72.1 | 59.5 |
| Proportion women who received fertility examination and/or ART | | | | | |
| Primary infertility | N = 206 | N = 192 | N = 111 | N = 36 | N = 17 |
| | % | % | % | % | % |
| No fertility examination | 21.4 | 18.2 | 25.2 | 36.1 | 58.8 |
| Fertility examination only | 10.2 | 22.9 | 27.0 | 38.9 | 29.4 |
| Fertility examination and ART | 68.4 | 58.9 | 47.7 | 25.0 | 11.8 |
| Secondary infertility | N = 338 | N = 248 | N = 176 | N = 93 | N = 25 |
| | % | % | % | % | % |
| No fertility examination | 46.7 | 46.8 | 46.6 | 48.4 | 80.0 |
| Fertility examination only | 15.1 | 21.0 | 31.8 | 32.3 | 16.0 |
| Fertility examination and ART | 38.2 | 32.3 | 21.6 | 19.4 | 4.0 |

reported either fertility examination (10.2%) or ART (68.4%), declining linearly to 41.2% (11.8%/29.4%) in the oldest cohort ($x^2$-trend, p<0.001) (Table 3, mid panel). Compared to women with primary infertility, a significantly lower proportion of women with secondary infertility received ART, and this proportion declined linearly from the youngest to the oldest birth cohort ($x^2$-trend, p<0.001) (Table 3, lower panel).

The "Baby take home rate" fell from nearly 60% among women with primary infertility in the 1966–75 cohort to 22% in the 1936–45 cohort (Table 4, upper panel). Most women with primary infertility and three or more children had given birth to twins and/or triplets. The "Baby take home rate" during ART for women with secondary infertility was lower across all birth cohorts when compared to those with primary infertility (Table 4, mid panel).

In total, 160 (50.3%) women with primary infertility had a child conceived during ART, compared to 102 (38.3%) women with secondary infertility. Most of these children were conceived during ART at the University hospital in Tromsø (82%, compared to 11% at other clinics in Norway and 7% abroad), with no difference between women with primary and secondary infertility. Overall, children conceived during ART accounted for 3.3% (95% CI: 2.9–3.7) of population growth in the 1966–75 cohort, declining to 0.7% (95% CI: 0.48–0.92) in the 1946–55 cohort (Table 5).

As educational level differed between the cohorts (Table 1), analysis by fertility status and educational level were restricted to the cohorts born after 1955. In stratified analyses, women with ≥4 years of college/university education were significantly more often voluntarily childless (adjusted odds ratio (OR) 1.5; 95% CI: 1.2–1.9) or had primary infertility (adjusted OR 1.5; 95% CI: 1.2–2.0) than fertile women. There were no differences between fertile women and voluntarily childless or infertile women in the other educational categories (using vocational/upper secondary education as the reference). Women with primary/unknown educational level had significantly lower odds of reporting secondary infertility (adjusted OR 0.7; 95% CI: 0.5–0.9) in comparison with fertile women, with no differences across the other educational categories.

There were no differences in educational level among infertile women (primary and secondary infertility combined) with and without a fertility examination, nor among those who

**Table 4. Number of children conceived during assisted reproductive technology (ART) by type of infertility and naturally conceived children born among women with secondary infertility (%).**

|  | Birth cohorts | | | | |
|---|---|---|---|---|---|
|  | 1966–75 | 1956–65 | 1946–55 | 1936–45 | 1916–35 |
| Age (years) | 40–49 | 50–59 | 60–69 | 70–79 | 80–98 |
| Proportion of women who had children during ART |  |  |  |  |  |
| Primary infertility | N = 141 | N = 113 | N = 53 | N = 9 | N = 2 |
| Number of children conceived during ART | % | % | % | % | % |
| 0 | 41.8 | 50.4 | 62.3 | 77.8 | 100.0 |
| 1 | 22.7 | 31.0 | 22.6 | 0 | 0 |
| 2 | 29.1 | 14.2 | 9.4 | 22.2 | 0 |
| 3 | 5.0 | 4.4 | 3.8 | 0 | 0 |
| 4 | 0.7 | 0 | 0 | 0 | 0 |
| 5 | 0 | 0 | 1.9 | 0 | 0 |
| 6 | 0.7 | 0 | 0 | 0 | 0 |
| Secondary infertility | N = 129 | N = 80 | N = 38 | N = 18 | N = 1 |
| Number of children conceived during ART | % | % | % | % | % |
| 0 | 53.5 | 58.8 | 84.2 | 88.9 | 0 |
| 1 | 35.7 | 28.7 | 13.2 | 11.1 | 100.0 |
| 2 | 10.1 | 12.5 | 2.6 | 0 | 0 |
| 3 | 0.8 | 0 | 0 | 0 | 0 |
|  | N = 129 | N = 80 | N = 38 | N = 18 | N = 1 |
| Number of naturally conceived children | % | % | % | % | % |
| 1 | 55.8 | 66.3 | 47.4 | 50.0 | 100.0 |
| 2 | 34.9 | 22.5 | 28.9 | 27.8 | 0 |
| 3 | 7.8 | 11.3 | 15.8 | 22.2 | 0 |
| 4 | 1.6 | 0 | 5.3 | 0 | 0 |
| 8 | 0 | 0 | 2.6 | 0 | 0 |

did and did not receive ART. However, a borderline significantly higher proportion of infertile women in the 1966–75 cohort received ART compared to the 1956–65 cohort (adjusted OR 1.3; 95% CI: 0.99–1.7) (Table 3).

## Discussion

This cross-sectional study covered participants born over a 60-year period: 1916–75. The estimates for primary and secondary involuntary childlessness were remarkably stable across

**Table 5. The contribution of assisted reproductive technology (ART) to population growth (%).**

|  | Birth cohorts | | | | |
|---|---|---|---|---|---|
|  | 1966–75 | 1956–65 | 1946–55 | 1936–45 | 1916–35 |
| Age (years) | 40–49 | 50–59 | 60–69 | 70–79 | 80–98 |
|  | N = 3373 | N = 3344 | N = 2675 | N = 1360 | N = 412 |
|  | N | N | N | N | N |
| Total number of children | 6605 | 6329 | 5576 | 3402 | 1164 |
| Children conceived during ART | 220 | 125 | 40 | 6 | 1 |
|  | % | % | % | % | % |
| Contribution to population growth | 3.3 | 2.0 | 0.7 | 0.18 | 0.09 |
| 95% confidence intervals | 2.9–3.7 | 1.7–2.4 | 0.48–0.92 | 0.04–0.32 | -0.08–0.26 |

cohorts. The incidence of primary involuntary childlessness was significantly higher among the 1956–75 cohorts than the 1916–55 cohorts, whereas the youngest birth cohort (1966–75) had a significantly higher incidence of secondary infertility than the older cohorts did. Across birth cohorts, the proportion of women with primary and secondary infertility remained stable, with a 40%/60% distribution. Nearly 80% of women with primary infertility in the 1966–75 cohort had a fertility examination, and nearly 70% received ART, decreasing linearly to 41% and 12%, respectively, in the oldest birth cohort. The same pattern was observed for secondary infertility, but at a lower magnitude. The "Baby take home" rate among women with primary infertility fell linearly from nearly 60% to 22% from the 1966–75 to the 1936–45 cohort; the corresponding fall was from 46% to 11% among women with secondary infertility. Women with the highest educational level were more often voluntarily childless and had primary infertility, whereas women with the lowest educational level had a lower prevalence of secondary infertility. There were no differences in fertility examinations or ART among infertile women by educational level.

The possibility to compare these results with those of other surveys on infertility depends upon response rates, definitions of infertility, and the age of respondents. For surveys of respondents aged 40 years and older, like the present study, estimates of infertility and use of health services may be considered permanent estimates. In the following discussion, we compare the cohort categories employed in the present study with overlapping cohorts in the literature.

Our observed prevalence of primary infertility (4.1%; 95% CI: 2.2–6.1) among women born before 1936 is like the prevalence (4.0%) reported from a random sample of Norwegian women born in 1934–37, when they were 40–44 years of age, but our observed secondary infertility rate (6.1%; 95% CI: 3.8–8.4) is lower than that reported in the 1977 (14%) survey [11]. Furthermore, our results underline the external validity of the 1936–45 cohort (primary infertility: 2.6%; 95% CI: 1.8–3.5; secondary infertility: 6.8%; 95% CI: 4.8.7.4) when compared to a random sample of Norwegian women aged 40–44 years in 1988 (primary infertility: 2%; secondary infertility: 13%) [11] and to a convenience sample from Norway (primary infertility 3.2%; secondary infertility: 6%) for the birth cohort 1946–55 [12].

Using a 24-month definition of infertility, a study from Scotland (Aberdeen) reported a higher prevalence of primary infertility (defined as no live birth) in the 1946–55 and 1956–65 cohorts than in our study, whereas rates of secondary infertility were similar [13]. Another Scottish study from the Grampian region reported 24-month prevalence of primary (defined as no live birth) and secondary infertility for the 1956–65 cohort (46–50 years) that were like our rates, though ours are based on a 12-month definition [14]. A study from the UK found the same prevalence of primary infertility (defined as no live birth) among women trying to conceive in the 1946–55 cohort as in our study, but they found a lower incidence for women in the 1956–65 cohort (4.2%; 95% CI: 3.8–4.8) [15]. Another UK study that used a 12-month definition found an 11–13% rate of subfertility for women (both primary and secondary infertility) in the 1936–65 cohort (aged 45–74 years at data collection in 2010–12). Using the same definition of subfertility, a Finnish study reported higher rates of subfertility than the UK study in cohorts born in the 1930s and 1940s [16].

We observed increasing proportions of women reporting fertility examination and ART from the oldest to the youngest birth cohorts, and higher proportions among women with primary than secondary infertility. The Scottish studies found minor differences in care-seeking behavior by type of infertility [13, 14], but these rates were lower than those we observed for women with primary infertility for the 1946–55 and 1956–65 cohorts, and higher for women with secondary infertility in both cohorts [13, 14]. As in our study, an increasing proportion of British women born between 1946–55 and 1956–65 consulted a physician for problems

conceiving, and about 50% of those who consulted received ART [15]. Our study found that increasing proportions of women received ART, close to 70% for primary infertility and 40% for secondary infertility in the 1966–75 cohort. This contrasts a record linkage study of medical treatment registry data and survey data from Finland, which reported higher proportions of subfertility, but lower treatment rates, in younger cohorts than older ones [16].

Whereas women born before 1936 received hardly any ART or reported any ART success, this success increased by birth cohort, reaching nearly 60% and 40% for women treated for primary and secondary infertility, respectively, in the youngest birth cohort (1966–75). The treatment options for the oldest cohort treated in the 1960s and 1970s were general limited to surgery and insemination, though some received ovulation stimulation, which was just being introduced. For the 1936–45 cohort, options for insemination and ovulation stimulation increased, and these women were the first to receive IVF in Tromsø and elsewhere. ART success for the remaining younger cohorts was related to new treatment options, and advances in and refinement of existing approaches related to ovulation stimulation; egg and sperm preparation; embryo culture, evaluation/selection, and transfer; embryos freezing, preservation, and thawing techniques; and other novelties.

For the cohorts born before 1956, ART success had minor impact on population growth in Tromsø. However, for the 1956–65 and 1966–75 cohorts, children conceived during ART comprised 2.0 and 3.3% of the total number of children born. Since 1997, the European IVF-monitoring Consortium has collected, analyzed, and reported aggregated data on ART from national registries, clinics, or professional societies. Since reporting started, the number of children conceived during ART has increased from 35 000 to over 195 000 in Europe. For some countries where services are easily accessible, children conceived during ART comprised 5–6% of all children born in 2016 [17]. Today, couples are delaying childbearing, thus many women become pregnant later in life, driving the need for ART. Indeed, in their projections of ART use, and what it will add to population growth worldwide, researchers have calculated that people conceived during ART may comprise as much as 1.4% to 3.5% of the global population by the year 2100 [18].

The proportion of voluntarily childless women increased from 5–6% for the cohorts born before 1956, to 9–10% for the cohorts born after 1955. The CIs for our estimates overlap with estimates reported for voluntarily childless women born before 1956 both in Norway [11, 12] and Scotland [13], but they are higher than those reported for the 1956–65 cohort in Scotland [14]. Our estimates for childlessness among women (number of children = 0, Table 1) reflects the national statistics for cohorts born after 1956 in Norway [19, 20], but they are lower than estimates for all cohorts born after 1939 in Finland, where the proportion of childless women at age 40 increased from 15% in the 1940–44 cohort to 20% for cohorts born after 1955 [21]. In Finland, childlessness was more common in highly educated women in older cohorts, but childlessness is now more often observed in low- and medium-educated Finnish women in younger cohorts. This in contrast to our findings, which showed that more highly educated women from the 1956–75 cohorts were voluntarily childless and more often had primary infertility.

Some strengths of Tromsø7 are its population-based approach and high response rate for all age groups. Moreover, we were able to run consistency analyses on the variable clinical infertility (infertility period of >1 year) across two surveys 8–9 years apart for 43% of the respondents. Reassuringly, 98.3% of women who reported no fertility problems in Tromsø6 gave the same response in Tromsø7. As we chose to base "infertility experience" on several variables from Tromsø7, we could not run consistency analyses in the same way, but 80% of infertile women answered "yes" to the question "Have you tried to conceive for more than 1 year without succeeding in becoming pregnant?" in both surveys. For women over 80 years of age

(born before 1936), the response rate was lower. However, we had national data from two studies for the 1916–36 and the 1936–45 cohort. Women were interviewed at age 40–44 years and reported a similar incidence of primary infertility, but our incidence of secondary infertility was like that in one of these studies [12], and lower than that observed in the other study [11], providing external validity to our results. One limitation is that our study did not have information on the timing of infertility experience, whether it was before the woman's first childbirth, between childbirths, or after their last childbirth. Therefore, secondary infertility also comprises periodic infertility, both before and after first childbirth, as our questionnaire did not differentiate between these entities. This may have underestimated our rate of primary involuntary childlessness across all cohorts, despite high internal and external validity, and overestimated the incidence of secondary involuntary childlessness.

## Conclusion

The incidence of primary involuntary childlessness increased from nearly 4% for women born in 1916–55 to 6% for the 1956–75 cohorts. The incidence of secondary infertility was higher than that of primary involuntary childlessness for all cohorts, reaching 10% for women born in 1966–75 relative to 6–7% for the older cohorts, and remained stable in a 60/40 proportion across cohorts. An increasing proportion of women from the oldest to the youngest cohort required fertility examination and ART, and this proportion was higher for women with primary than secondary infertility. As ART successes were negligible for cohorts born before 1956, advances in ART over the past 50 years comprised 2.0% and 3.3% of population growth for the 1956–65 and 1966–75 cohorts, respectively.

## Author Contributions

**Conceptualization:** Finn Egil Skjeldestad.

**Data curation:** Finn Egil Skjeldestad.

**Formal analysis:** Finn Egil Skjeldestad.

**Investigation:** Finn Egil Skjeldestad.

**Methodology:** Finn Egil Skjeldestad.

**Project administration:** Finn Egil Skjeldestad.

**Validation:** Finn Egil Skjeldestad.

**Writing – original draft:** Finn Egil Skjeldestad.

**Writing – review & editing:** Finn Egil Skjeldestad.

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
