## [Decision Letter · Decision Letter 0]

14 Mar 2023

PONE-D-23-03364Minor changes in incidence of primary and secondary infertility across birth cohorts 1916 to 1975, but major differences in treatment successPLOS ONE

Dear Dr. Skjeldestad,

Thank you for submitting your manuscript to PLOS ONE. After careful consideration, we feel that it has merit but does not fully meet PLOS ONE’s publication criteria as it currently stands. Therefore, we invite you to submit a revised version of the manuscript that addresses the points raised during the review process.

We look forward to receiving your revised manuscript.

Kind regards,

Jacopo Di Giuseppe

Academic Editor

PLOS ONE

Journal Requirements:

   "No funding"

Reviewers' comments:

Reviewer's Responses to Questions

**Comments to the Author**

1. Is the manuscript technically sound, and do the data support the conclusions?

Reviewer #1: No

Reviewer #2: Yes

Reviewer #3: Partly

2. Has the statistical analysis been performed appropriately and rigorously? 

Reviewer #1: Yes

Reviewer #2: Yes

Reviewer #3: No

3. Have the authors made all data underlying the findings in their manuscript fully available?

Reviewer #1: No

Reviewer #2: No

Reviewer #3: No

4. Is the manuscript presented in an intelligible fashion and written in standard English?

Reviewer #1: Yes

Reviewer #2: Yes

Reviewer #3: No

5. Review Comments to the Author

Reviewer #1: The title: Seems interesting but needs to be rewritten, please make it concise.

Abstract: No new idea, the statements are cut and paste from the main body of the study.

Introduction: Emphasizes the history. The transition from the history to the objective is abrupt. There seems to be no smooth flow of ideas (disconnected).

The manuscript is not complex, i.e. since it is cross sectional study, with the clearly defined cohorts.

The grammar is also acceptable.

The only concern I have is the knowledge it adds to the existing understanding of infertility and its management. The internal validity seems acceptable, but the reproducibility, the generalization, the application is the limitation. What is the main idea the investigators want to address. What was the gap identified, statement of the problem? What is the significance of this study?

The statements in most parts of the manuscript are duplications (repetitions) of same statements.

Though the grammar usage seems acceptable the flow of idea does not seem attractive to the reader (my perspective). The statistics used were appropriate to the proposed design (internal validity).

The strength and the limitation of the study is not addressed.

The result section is well summarized.

The discussion section: I am not clear how the references used in the discussion section much the contents of the manuscript in the current study. In addition, the discussion section does not explain why or how the similarity or the differences between the results of the current study and the results in the refences used. This needs major revision.

The conclusion section is the repetition of the some paragraphs in the discussion (cut and paste).

Reviewer #2: The manuscript is technically sound and the data supports the conclusions drawn from the study.

2. However, the fact that births were used to determine fertility rather than pregnancies should have been indicated as a major limitation of the study. The Author did state that they did not have information on pregnancies before childbirth, between childbirth and after childbirth. This affected the definition of secondary infertility thus making comparisons with existing studies difficult.

3. The following minor typographical and grammatical errors were observed:

a. ABSTRACT: Line 10: Women with secondary infertility were those who reported infertility experience and at had least one naturally conceived child…. To read “had at least”.

b. Page 2 Line 34: “defined as reporting one of more of the following” ………To read: “one or more”.

c. Page 2 Line 36: …..”experience and at had least one” …. To read…….. “had at least.

d. Page 5 Line 96: “one of more” … To read “one or more”..

e. Page 13 Line 265: “ART success for the remaining cohorts benefited from new treatment options”…. ! It is not clear what the Author is saying here!!! Could it be that “ART success for the remaining cohorts was related to new treatment options”, and advances in and refinements of existing approaches…………… Review please.

f. Page 14 Line 295: “Have you to tried to” … To read “Have you tried to”…

g. Page 314 Line 300: “was alike” … to read “was similar”

h. Page 14 Line 306: delete “10%” after 1966-75.

i. Page 15 Reference 2: “chaperon” to read “Chaperon”

4. I recommend the that the Author should address the issues raised and effect the corrections before acceptance for publication, please.

Reviewer #3: Thank you for your best work. However the paper iis long and has not focused on evolution of ART within the years understudy. There is need also to edit the grammar in the entire manuscript, Sample size determination and study design is not stated. Result presentation is crowded and discussions have not been robust. Please also include study limitation. What is the take home message from this study?

6. PLOS authors have the option to publish the peer review history of their article (what does this mean?). If published, this will include your full peer review and any attached files.

Reviewer #1: No

Reviewer #2: **Yes: **Dr Francis E. Alu

Consultant Obstetrician & Gynaecologist

Abuja Nigeria

Reviewer #3: No

---

## [Author Response · Author response to Decision Letter 0]

1 May 2023

Reviewer #1: 

The title: Seems interesting but needs to be rewritten, please make it concise.

Response: I have changed the title. The word infertility is replaced with involuntary childlessness. The title reflects the results. “Minor changes in incidence of primary and secondary involuntary childlessness across birth cohorts 1916 to 1975, but major differences in treatment success”. In my opinion this is a concise result driven title. However, I can easily change the title to a more general title on the editor’s request.

Abstract: No new idea, the statements are cut and paste from the main body of the study.

Response: Abstracts summarize aims, methods, results ending with a conclusion. The language reflects the writing in the main manuscript.

Introduction: Emphasizes the history. The transition from the history to the objective is abrupt. There seems to be no smooth flow of ideas (disconnected).

Response: Study participants have had their fertility careers across the evolution of modern assisted reproductive technologies. That is why the introduction is made short from ancient history to what started in the 1960s, and its rapid evolution of the different treatment modalities. I agree it is “abrupt,” and there is a flow information that describe the evolution, but in telegram style, short and rousing. I have not changed the introduction.

The manuscript is not complex, i.e. since it is cross sectional study, with the clearly defined cohorts.

The grammar is also acceptable.

Response: Thanks for the positive feedback. The grammar has been edited by an international “editing firm.”

The only concern I have is the knowledge it adds to the existing understanding of infertility and its management. 

The internal validity seems acceptable, but the reproducibility, the generalization, the application is the limitation. 

Response: I consider these statements as the reviewer’s assessments on overall quality. I am glad the reviewer found the internal validity “acceptable.” However, I am not concerned about the reproducibility and generalization of the results. All over the world there has been tremendous achievements in ART success. This study shows how this changed over cohorts without any particular treatment options, to those cohort that received increasingly refined treatments, with high success rates. 

What is the main idea the investigators want to address. What was the gap identified, statement of the problem? What is the significance of this study?

Response: Data presented assessed primary and secondary involuntary childlessness, the need for infertility work-up, and outcome of infertility treatment across birth cohorts over a 60-year lifespan in a well-defined geographical population. As this reviewer states in an above sentence that “the internal validity seems acceptable,” but the “reproducibility and generalization” is the limitation. These data yields women residing in Tromsø city. These data may be reproduced in other settings with different or similar estimates. The generalization of these data is discussed considering what is published from other countries with good external validity to studies from Scotland and England which have similar health care systems as the Norwegian. 

The statements in most parts of the manuscript are duplications (repetitions) of same statements.

Response: I do not agree to this statement. The discussion part follows the result presentation which is the way of scientific writing. 

Though the grammar usage seems acceptable the flow of idea does not seem attractive to the reader (my perspective). 

Response: There is a clear flow of ideas in the result and discussion part; trends on primary and secondary infertility, need for infertility work-up, and outcome of treatment. These data are discussed in the same flow of data as conveyed in the results.

The statistics used were appropriate to the proposed design (internal validity).

Response: grateful for the positive feedback!

The strength and the limitation of the study is not addressed.

Response: Strengths and limitations were included in the first submitted manuscript; lines; 290-303. More text is added in the revised manuscript. 

The result section is well summarized.

Response: grateful for the positive feedback!

The discussion section: I am not clear how the references used in the discussion section much the contents of the manuscript in the current study. 

In addition, the discussion section does not explain why or how the similarity or the differences between the results of the current study and the results in the refences used. This needs major revision.

Response: After summarizing the study results in the first paragraph of the discussion, I wrote a paragraph of problems comparing results to other studies.

“The possibility to compare these results with those of other surveys on infertility depends upon response rates, definitions of infertility, and the age of respondents. For surveys of respondents aged 40 years and older, like the present study, estimates for infertility and use of health services may be considered permanent estimates. In the following discussion, we compare the cohort categories employed in the present study with overlapping cohorts in the literature. “ 

As it is difficult to find exact cohorts in the literature, I needed to compare our results with overlapping cohorts. For each study that our study is compared with I defined the comparing outcomes within timeline for infertility definition or whether “no births” or “no pregnancy” was the infertility “denominator”. In this way our comparisons are transparent to other studies. This is systematically written in the discussion. Some minor revisions are made. 

The conclusion section is the repetition of the some paragraphs in the discussion (cut and paste).

Response: The conclusion reflects the results. The discussion part resembles in a natural way the dissemination of the results. This is a very usual way of writing discussions/conclusions in any paper.

Reviewer #2: The manuscript is technically sound, and the data supports the conclusions drawn from the study.

Response: I appreciate this positive feedback

2. However, the fact that births were used to determine fertility rather than pregnancies should have been indicated as a major limitation of the study. The Author did state that they did not have information on pregnancies before childbirth, between childbirth and after childbirth. 

Response: I have in the revised manuscript defined primary infertility as synonymous with involuntary childlessness, respective secondary infertility as synonymous to secondary involuntary childlessness. This is explicitly written in several parts to remind the reader about the definition.

When our data is compared with other studies, the definition of infertility in other studies is added in the text, to make the comparisons transparent.

This affected the definition of secondary infertility thus making comparisons with existing studies difficult.

Response: Definitions for secondary infertility for comparative studies are provided in the text. As written in the limitations, primary involuntarily childlessness may be underestimated, and consequently secondary involuntarily childlessness overestimated.

3. The following minor typographical and grammatical errors were observed:

a. ABSTRACT: Line 10: Women with secondary infertility were those who reported infertility experience and at had least one naturally conceived child…. To read “had at least.”

Response: revised

b. Page 2 Line 34: “defined as reporting one of more of the following” ………To read: “one or more”.

Response: revised

c. Page 2 Line 36: ”experience and at had least one” …. To read…….. “had at least.

Response: revised

d. Page 5 Line 96: “one of more” … To read “one or more”..

Response: revised

e. Page 13 Line 265: “ART success for the remaining cohorts benefited from new treatment options”…. ! 

It is not clear what the Author is saying here!!! Could it be that “ART success for the remaining cohorts was related to new treatment options”, and advances in and refinements of existing approaches…………… Review please.

Response: revised in line with what the reviewer suggested.

f. Page 14 Line 295: “Have you to tried to” … To read “Have you tried to”…

Response: revised

g. Page 314 Line 300: “was alike” … to read “was similar”

Response: revised

h. Page 14 Line 306: delete “10%” after 1966-75.

Response: revised

i. Page 15 Reference 2: “chaperon” to read “Chaperon”

Response: revised

4. I recommend the that the Author should address the issues raised and effect the corrections before acceptance for publication, please.

Reviewer #3: Thank you for your best work. 

However, the paper is long and has not focused on evolution of ART within the years understudy. 

Response: the birth cohorts understudy has had their infertility periods from 1960 and onwards, which is in during the evolution of ART. However, as we can not exactly time when the hospital where the women were treated started the different treatment modalities, I can not relate the specific examinations and treatment provided. On the other hand, the treatment was provided at the University hospital of Northern Norway in Tromsø, elsewhere in Norway, and abroad. Therefore, we do not have data to provide more specific treatment modalities than provided in the result part.

There is need also to edit the grammar in the entire manuscript, 

Response: an international editing firm has assessed the grammar

Sample size determination and study design is not stated. 

Response: in health surveys sample size determination is not that usual. I have provided confidence intervals on major results, which reflects the “sample size” on the underlying populations. 

Result presentation is crowded and discussions have not been robust. 

Response: It is difficult to respond to this general subjective perspective of the reviewer. 

Please also include study limitation. 

Response: Strengths and limitations were included in the first submitted manuscript; lines; 290-303. More text is added in the revised manuscript. 

What is the take home message from this study?

Response: reflected in the title: “Minor changes in incidence of primary and secondary involuntarily childless across birth cohorts 1916 to 1975, but major differences in treatment success”.

---

## [Decision Letter · Decision Letter 1]

22 May 2023

PONE-D-23-03364R1Minor changes in incidence of primary and secondary involuntary childlessness across birth cohorts 1916 to 1975, but major differences in treatment successPLOS ONE

Dear Dr. Skjeldestad,

Thank you for submitting your manuscript to PLOS ONE. After careful consideration, we feel that it has merit but does not fully meet PLOS ONE’s publication criteria as it currently stands. Therefore, we invite you to submit a revised version of the manuscript that addresses the points raised during the review process.

We look forward to receiving your revised manuscript.

Kind regards,

Jacopo Di Giuseppe

Academic Editor

PLOS ONE

Journal Requirements:

Reviewers' comments:

Reviewer's Responses to Questions

**Comments to the Author**

Reviewer #2: All comments have been addressed

Reviewer #3: All comments have been addressed

2. Is the manuscript technically sound, and do the data support the conclusions?

Reviewer #2: Yes

Reviewer #3: Partly

3. Has the statistical analysis been performed appropriately and rigorously? 

Reviewer #2: Yes

Reviewer #3: No

4. Have the authors made all data underlying the findings in their manuscript fully available?

Reviewer #2: Yes

Reviewer #3: Yes

5. Is the manuscript presented in an intelligible fashion and written in standard English?

Reviewer #2: Yes

Reviewer #3: No

6. Review Comments to the Author

Reviewer #2: The authors have been able to address all the issues I raised and corrected the typographical and grammatical errors.

Reviewer #3: Thank you for interesting paper. However:

1. Sampling method and sample size need to be Stated

2. I think this is retrospective cross sectional study design

3. Any ethical approval or waiver?

4. English needs thorough editing because some sentences especially in the abstract are not making sense.

7. PLOS authors have the option to publish the peer review history of their article (what does this mean?). If published, this will include your full peer review and any attached files.

Reviewer #2: **Yes: **Dr Francis E. Alu, MD

Consultant Obstetrician & Gynaecologist

Abuja Nigeria

Reviewer #3: No

---

## [Editor Report · Decision Letter 2]

7 Jun 2023

Minor changes in the incidence of primary and secondary involuntary childlessness across birth cohorts 1916 to 1975, but major differences in treatment success

PONE-D-23-03364R2

Dear Dr. Skjeldestad,

We’re pleased to inform you that your manuscript has been judged scientifically suitable for publication and will be formally accepted for publication once it meets all outstanding technical requirements.

Kind regards,

Jacopo Di Giuseppe

Academic Editor

PLOS ONE

---

## [Editor Report · Acceptance letter]

19 Jun 2023

PONE-D-23-03364R2 

Minor changes in the incidence of primary and secondary involuntary childlessness across birth cohorts 1916 to 1975, but major differences in treatment success 

Dear Dr. Skjeldestad:

I'm pleased to inform you that your manuscript has been deemed suitable for publication in PLOS ONE. Congratulations! Your manuscript is now with our production department. 

Kind regards, 

on behalf of

MD Jacopo Di Giuseppe 

Academic Editor

PLOS ONE